# Conservation and Development: Reassessing the Florida 2070 Planning Project with Spatial Conservation Prioritization

**Fengze Lin, Mingjian Zhu * and Fengming Chen**

School of Design, South China University of Technology, Guangzhou 510006, China
* Correspondence: zhumj@scut.edu.cn

**Abstract:** The state of Florida is renowned for its globally recognized biodiversity richness, but it currently suffers from an ongoing population boom and corresponding urban sprawl resulting in the emergence of severe conservation conflicts, especially in southern parts of Florida. To mitigate the intense competing land use situation and comprehend the dynamic complex relationship between conservation and development, this study argues that both ecological and social dimensions should be taken into account for spatial analysis and underpin zoning decisions empirically in the phase of landscape planning. Choosing South Florida as the study site, we implemented focal-species-based spatial conservation prioritization analysis using Zonation software to identify the highest priority areas and accordingly evaluate two varying land use scenarios provided by the Florida 2070 Project. From a novel perspective of impact avoidance, the inverse prioritization method was applied in this study, intended to minimize negative human impacts and examine the effectiveness and suitability of Florida's future land use projections. After comparing and integrating social-ecological data through mapping, the study uncovered a holistic view of conservation conflicts in Florida and articulated trade-offs for all parties of the local ecosystem striving to reconcile human–wildlife conflicts in Florida and imply a sustainable win-win strategy in the stage of regional landscape planning.

**Keywords:** conservation conflict; spatial conservation prioritization; inverse prioritization; regional planning evaluation; zonation software

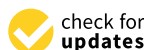



## 1. Introduction

Human population increases and corresponding unprecedented global biodiversity declines have escalated a looming social and environmental dual-crisis that may trigger dreadful consequences for human society as well as global and regional ecosystems [1–4]. In the past few decades, the framework or paradigm of human–wildlife conflict has been extensively applied by scholars and relevant literature has increased considerably [5], nonetheless inherently addressing the opposing positions of we human beings and nature [6]. The seeming dichotomy between development and conservation ignores genuine, intertwined sophisticated relationships and interactions between humans and nature [6]. Thus it is a more rational and arguable viewpoint to recognize two parties of conflict as a whole. Humans can take conservation actions to reconcile conservation conflicts rather than resolving or eliminating them [7]. To balance society development and biodiversity conservation and accomplish a 'win-win' situation for humans and nature [7,8], it is necessary to identify underlying objectives held in both ecological and social dimensions, especially for the commonly imperceptible human–human conflicts that interactively shape social values [9].

Conflict reconciliation involves processes in multiple aspects, including conflict mapping, conflict planning, demonstrations of strategies and evaluation of outcomes [10], among which spatial conservation prioritization (SCP) is an effective land-use planning tool that can be capable of the evaluating biodiversity status of specific regions in a balanced approach and utilized for mitigating conflicts in advance [2,11,12], through specific

measures such as ranking for high-priority areas, identifying green corridors, ecological network delineation, allocation of habitat restoration and offsetting distribution [13–16]. Underpinned by previous landscape planning studies which employed traditional conservation prioritization methods [2,17–19], their contributions are indisputable in informing ecological protection efforts with transparent trade-offs [17,20], but these studies neglected the flip side of urban planning zoning issues, which are precisely the non-biodiversity or human-dimensional aspects [8,14,17,21]. These social dimensions should be considered, along with interests in biodiversity conservation in the policy-making stage. Once the natural and social science factors are connected [22,23], assessment of human–animal conflicts [24] can assist urban planners to explore trade-offs effectively and explicitly from a more comprehensive perspective [8,25].

With highly urbanized areas and ecologically-valuable reserves, the sunshine state urgently requires effective conservation planning solutions to manage and diminish continued human–wildlife conflicts under the background of continuous biodiversity loss, future sea level rise scenarios and competing land use [19,26,27]. Florida's severe human–animal conflicts are provoked by the tension between its exceptional position as a harbor for many global biodiversity hotspots [28,29] and decades of high-speed population increase [30]. Within such a particular context, the Florida 2070 joint project was collectively implemented by local governors and academic research groups intending to explore distinct developmental pathways to accommodate projected human population growth by 2070 [31,32]. In the Florida 2070 projections, two different scenarios (Trend and Alternative) were generated based on varying human development assumptions, where one follows the present inefficient urban sprawl pattern and the other stands for the compact development approach. Although these two variable projections are hypothetical and with stochasticity, they can certainly be comprehended as a synthesis of general human development preferences [8], or, in other words, crucial social science evidence for potential land-use planning analysis.

The Alternative 2070 scenario suggests a land-use planning method using the 'compact city approach' [33] by assuming a new population for redevelopment or infill development [34]. Compact land-use pattern aims to spare considerable greenfield sites for conservation purposes, which has similar outcomes to another conservation planning application, impact avoidance [16,33]. As an efficient environmental zoning approach, impact avoidance can be easily achieved by techniques of inverse spatial conservation prioritization [13,35]. In this study, we took advantage of this robust approach to make our SCP identification and succeeding analysis more complete.

This paper uniquely combined scientific evidence from the human and nature conservation sides by reassessing Florida's future development projections with a two-sided spatial conservation prioritization analysis method to help acknowledge a holistic view of conservation conflicts in Florida. Linking impact avoidance with conflict reconciliation within the study area of South Florida, our approach identified and analyzed highest and lowest priority areas and integrated with different land use patterns to reveal intractable development–conservation conflicts early in the planning phase and seek a durable win-win scenario for all parties of the ecosystem. As long as relevant multi-dimensional data are available, we encourage future conservationists and decision-makers elsewhere to adapt our reassessment methods and contribute to sustainable regional planning strategies to minimize further biodiversity loss.

## 2. Materials and Methods

### 2.1. Study Area

Florida hosts the most significant biodiversity among all U.S. states and has more than 16,000 kinds of native species found nowhere else on the planet [27]. However, the sunshine state suffers the most from booming population growth and aggressive urban sprawl. Unlike most existing literature on Florida conservation, our studies focused on South Florida to address land-use issues and human–wildlife conflict, which are common

global problems. South Florida distinguishes itself from other ecoregions in Florida by its uniqueness in natural and human-dimensional factors.

The southern region of Florida, our study site (Figure 1), covers 55,604.4 km² and ranges from the upper Lake Okeechobee to the end of the Florida Peninsula along with Florida Keys islands, consisting of 16 counties: Broward, Charlotte, Collier, Desoto, Glades, Highlands, Hendry, Indian River, Lee, Martin, Miami-Dade, Monroe, Okeechobee, Palm Beach, Sarasota and St. Lucie. South Florida mainly encompasses subtropical and tropical climate zones, with annual average precipitation of 59 inches and temperatures ranging from 47° to 90° F [36]. The study area of our assessment involves 4 substantial National Parks administered by the National Park Service (NPS) and 70 wildlife management areas (WMA) preserved by the Florida Fish and Wildlife Conservation Commission (FFWCC), respectively taking up 91.43% of National Park area and 36.3% WMA (Florida Geographic Data Library). Many unique ecosystems and corresponding wildlife habitats are located within these protected areas in South Florida, such as Lake Okeechobee, the Everglades Wetland, the Big Cypress Swamp and the Florida Keys islands [37]. With such a robust ability to support an amazingly diverse group of flora and fauna, our study area should undeniably receive recognition as one of the most vital biodiversity hot spots statewide and be among the highest global rankings for ecological conservation [27,38,39]. The species richness of our study areas has not been quantified and collected by reliable literature. However, there are more than 68 threatened animal species (USFWS 1999) merely within the Greater Everglades ecosystem that comprehend a total of 1033 plant taxa, 59 reptile taxa, 76 mammal taxa, 432 fish taxa, 349 bird taxa, 38 amphibian taxa and 459 bird taxa [37].

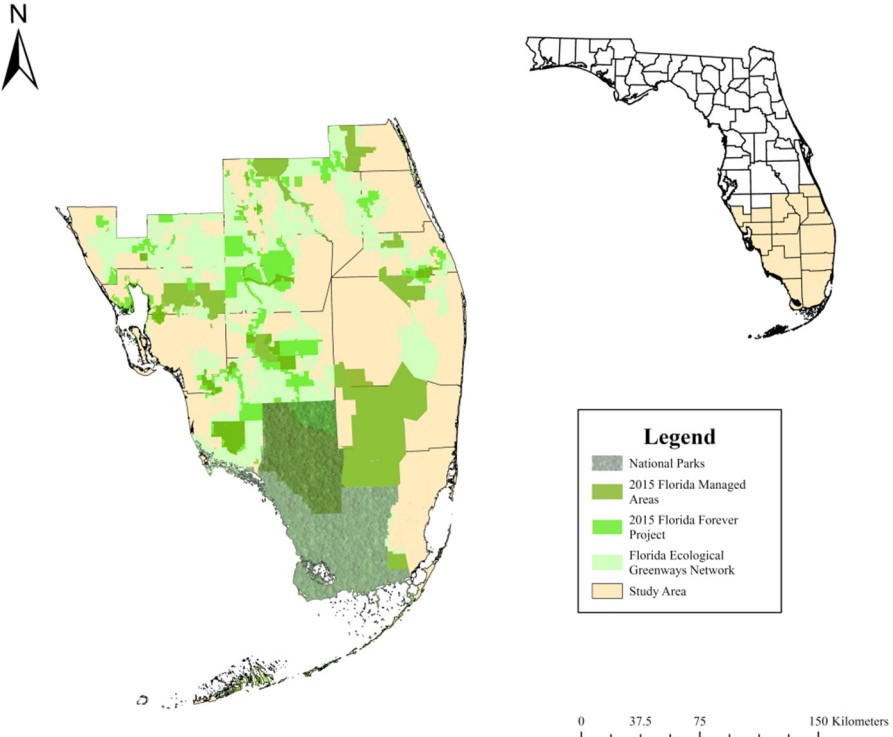

**Figure 1.** Study area.

Even with large areas of agriculture and other types of greenfield sites, rapid urban development and population growth in recent years have collectively put unprecedented pressure on Florida, especially in the South region [34,36,40]. Historically, the population of our study area has increased speedily from 6,092,509 to 8,852,679 during the period of 30 years between 1990 and 2020, with an incredible percentage increase of 45.3% [32,41]. Nearly 9 million (8,959,286) people currently live in this area of great wealth in 2021 [42] and it is credibly predicted that over 4 million people will be added by 2070 with a growth

rate of 50.12% as stated in the Population Projection of FDOT (Florida Department of Transportation) [32]. In line with the Florida 2070 report [34], this region is expected to see not just a population explosion but also intensive land resources exploitation that the former relies on. Aside from this, it is noteworthy that an extremely uneven population distribution pattern occurs in our study area, where highly urbanized areas such as Broward (2), Miami-Dade (6), Lee (8) and Sarasota (9) rank among the top 10 most dense counties statewide whereas Glades (66) and Hendry (56) are occupied mainly by reserve areas and agricultural land reasonably supporting the least population per square mile [41].

In the Southern region, a high percentage of wildlife habitat reserves and agricultural land are deemed crucial to Florida's ecosystem, which conflicts with the extensive metropolitan area, high population densities and future population rise [30]. Such a tug of war significantly influenced two projections of Florida 2070 and provoked our further conservation-wise assessment of current population-based simulations. Synthesizing the ecological importance of natural communities and social urban development cell needs, we want to focus on the dynamic relationship between conservation and development and thus find South Florida compelling in reassessing the legitimacy of Florida's existing development and conservation trajectory, as well as explicating and disclosing severe human–wildlife conflicts within Florida and even worldwide.

### 2.2. Framework of Reassessment Procedures

With the goal of resolving conservation conflict in Florida, available data sets of the study area and advanced quantitative planning tools were preferred as our pieces of evidence and methods underpinning advanced studies. In order to clarify and confirm the following research procedures, we articulate a reassessment framework for our study in Figure 2, including five successive stages: (1) collection of land cover data and Florida 2070 Project data as social science evidence; (2) acquisition of biodiversity distribution data from listed focal species in our study site as nature-side data; (3) using a quantitative computational planning approach, Zonation, to process species-based data, then conducting (inverse) spatial conservation prioritization; (4) comparison between Trend/Alternative development plan with the Top/Lowest ranking maps and close-up analysis for synthesizing two-sided plannings; (5) further discussion and justification of achieved work along with a future vision for future implementation.

### 2.3. Zonation 4 Analysis for the Florida 2070 Project Scenarios

In this paper, we implemented the spatial prioritization for the study area's current conservation status using the Zonation 4 software [11], with Florida 2070 projection maps and focal species habitat map as input files. The features and strength of Zonation software are that it can generate complementarity-driven conservation ranking of the landscape by literately removing grid cells in various rules of conservation value aggregation and tries to maintain a balanced coverage of all input biodiversity components throughout the ranking to ensure the complementary balance between different species.

Technically, Zonation 4 software provides four options for aggregating conservation value that determines the removal order of cells and how the balance between features is implemented, while conducting priority ranking: (1) core-area Zonation (CAZ), (2) additive benefit function (ABF), (3) target-based benefit function, and (4) the generalized benefit function. In our case, we employed the additive benefit function method of ranking because the ABF rule tends to produce a more balanced feature distribution map without bias for the highly-weighted features, compared to core-area Zonation [43]. The ABF cell removal method is appropriate for our study area of South Florida, which needs to be equally evaluated as a whole, as it takes into account all biodiversity feature proportions in a given cell rather than focusing on a single feature that has the highest, as core-area Zonation rule does [43].

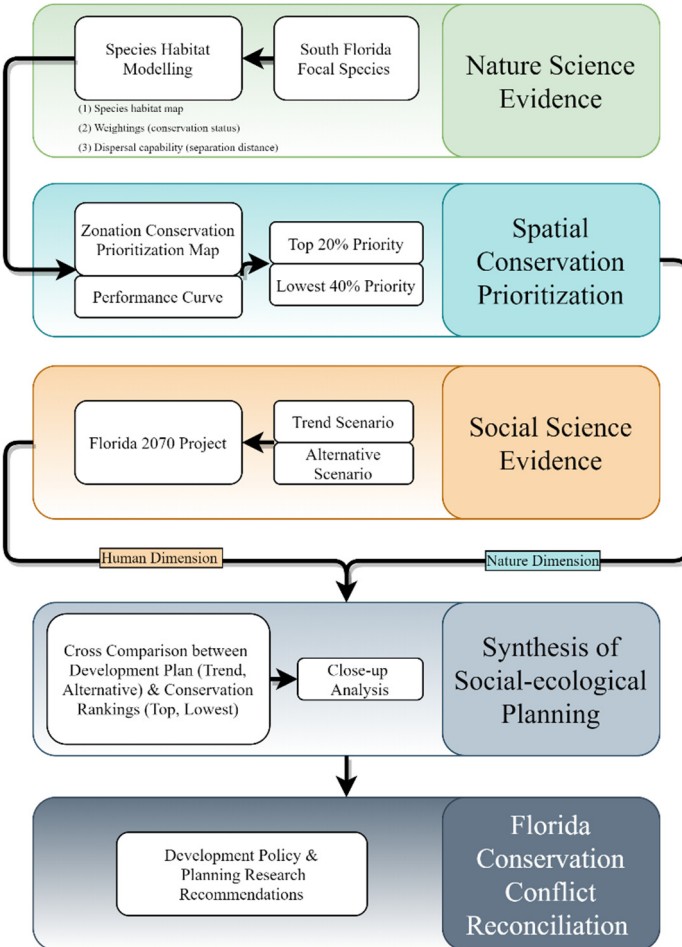

**Figure 2.** Reassessment Procedure.

To conduct a Zonation analysis, there are some compulsory input files needed for further software processing, including (a) a biodiversity feature distribution map (59 species' distribution map in South Florida for this study) as a GIS raster file; (b) a run settings file that contains all basic Zonation settings; (c) a biodiversity feature list file that contains a list of selected species together with their parameters, including species weightings and α value of the biodiversity feature-specific scale of landscape use (dispersal capability or the home range sizes of species for this study). Precisely for this study, we acquire reliable species distribution maps from the United States Geological Survey (USGS) as the biodiversity feature map. Endangered species is one of the most influential factors considered in conservation planning analysis [44]. Therefore, this study's 59 focal species of the South Florida region were selected based on (1) expert opinions from Southwest Florida Conservation Design [41], (2) FFWCC Technical Report (2009) [45] and (3) FFWCC Florida Endangered and Threatened Species List (2022) [46]. The chosen 59 endangered species are composed of 34 birds, 13 reptiles, 11 mammals and 1 amphibian. Distribution maps of these 59 species from USGS were included in the biodiversity feature list file and different weightings for involved species were referred to their global and state conservation status on NatureServe Explorer, as well as the Florida Natural Areas Inventory (FNAI) (see Table A1). The α value, which indicates dispersal capability, separation distance, or home range sizes of species, should always be specified. In our case, the α values were calculated based on each of the 59 species' average dispersal capability, sourcing from convincing expert knowledge from NatureServe Explorer and FFWCC (see Table A2). There is a need to justify that separation distances are estimated by wildlife occurrences, while occurrences are of practical size for conservation purposes and do not necessarily represent discrete populations or metapopulations. The α value of certain species should be deemed as a

reasonably compromised parameter between the structural requirement for conservation analysis and these birds' high mobilities in nature.

The output files of a Zonation analysis include one prioritization ranking map generated by the cell removal algorithm regarding inputted features (species distribution in our case), as well as one feature-specific representation loss curve/performance curve [43]. The prioritization map and curve are two intuitive visual representations of local conservation planning analysis, which also unfold quantitative relationship between viability of 59 focal species and simulated overall landscape ecological performance.

### 2.4. Assessing Florida 2070 Development Scenarios

Florida 2070 is a collective project conducted by the Florida Department of Agriculture and Consumer Services, the University of Florida GeoPlan Center and 1000 Friends of Florida. The Florida Bureau of Economic and Business Research (BEBR) conservatively estimates that Florida will accommodate about 15 million new residents in 50 years [42], which becomes the engine of succeeding mappings. Apart from BEBR population projections, the actual Florida 2010 development distribution map (Baseline 2010) serves as the foundation for upcoming development scenarios.

In order to cope with expected population growth and corresponding exploitation of limited land resources, the project aims to present and visualize the state's potential development challenges with varying solutions, respectively represented by two land-use scenarios: 2070 Trend and 2070 Alternative. The former planning follows current development strategies to accommodate new residents, which suggests an easy but inefficient land-use model without acknowledging the significance of green space and animals dependent on it [31]. In comparison, the alternative projection stimulates a more sustainable pathway where parts of the joining population will be allocated to existing urban areas while ensuring wildlife's viability and persistence through adequate protection methods for conservation areas.

Florida Trend 2070 and Alternative 2070 are distinguished inherently by the technical simulation assumptions below. The same suitability criteria are established and shared for two development plans, considering factors such as ongoing regional planning programs, availability of natural resources and conditions of or proximity to urban infrastructures. The Trend scenario only follows the current extensive development pattern, distributing new population outside of existing urban areas and possibly allocating new population to current agricultural lands. Nevertheless, the Alternative scenario assumes measurable proportions of the new population for infill development or redevelopment [34] and a 20% increase in gross development densities over the Trend scenario. Especially in the Alternative scenario, our study areas' mean redevelopment percentage is less than 24% (ranging from 10% to 60%). Therefore, limited proportions of the new population can be accommodated within South Florida, unfolding competing land-use situations in our study site. In terms of the protected lands, no new conservation land would be protected in the Trend projection, whereas the Alternative scenario is expected to reserve more greenfield sites, including the 2015 Florida Forever Project Areas, 2015 Florida Managed Areas and Florida Ecological Greenways Network Priorities 1 & 2 [31].

The Florida 2070 Project provides transparent mapping files publicly and we obtained relevant land use datasets (2010 Baseline, 2070 Trend, 2070 Alternative) from the Florida Geographic Data Library (FGDL). The land use map is divided into five categories: developed, agriculture, protected not agriculture, protected agriculture and others. Specifically, the scope of developed areas includes buildings, roads, interstates and vacant platted parcels [34].

The 2070 Trend scenario persists with a business-as-usual development pattern [3] while the 2070 Alternative scenario designs a compact and durable urban planning solution. Based on the above resources and the study site of South Florida, the very first step in reassessment of the 2070 project was to identify ecological core areas or high-richness habitats (top priority sites) and greenfield sites that have the potential to be exploited at a low eco-

logical cost (lowest priority areas) according to focal-species-based Zonation prioritization results. Secondly, we focused on GIS-driven overlapping and further evaluated the future land use scenarios with Zonation prioritization rankings directly and inversely. The Florida 2070 project was solely motivated by population growth and the need to combine with biodiversity conservation identification outcomes for resolving human–wildlife conflicts. Based on the focal-species-based ecological prioritization results (top 20% and lowest 40%), relevant categories of land changes were discussed under Trend and Alternative scenarios. Regarding the top 20% high-priority areas, comparisons of developed areas colliding with top 20% priority and top 20% greenfield sites were conducted to address the discrepancy between the two scenarios; whereas, from the view of impact avoidance, the two scenarios' varying composition of related land was revealed, which includes areas improperly developed, suitable areas for development and potential areas for future development. Apart from statistical comparisons, we dived deep into small-scale regions and presented close-up case studies to directly elaborate and emphasize the spatial contrast of the two projections. Insightfully, this study particularly examined land use distribution shifts in the past 10 years, showing a gap between 2070 scenarios and the current rapid urbanization trend and emphasizing the necessity and urgency for balanced conversation planning.

## 3. Results

### 3.1. Landscape Prioritization with Zonation 4 Software

As Figure 3 shows, mapped at a 30 m resolution, the priority ranking map of the study area in southern parts of Florida was generated by Zonation 4 software with a color gradation symbology indicating the prioritized ecological value from low to high and zero to one, with the current reserve areas overlaid and displaying as black hatching cells. Considerations should be given to high-priority areas as well as ecologically low-richness sites. Most high-priority areas are distributed along the southwestern coastline and the Florida Keys islands, which contain several existing critical protected areas. In addition, some medium to high-priority patches remain at the urban fringe of the central-north parts of the study area and even overlap with dense metropolitan areas. Referring to hatching cells of Florida's current conservation areas on the map, it is evident that managed reserve areas already covered most of the high-priority areas. On the other hand, low-biodiversity-feature lands include city impermeable surfaces and residential areas. Thus, the highly urbanized east coast generally receives the least spatial prioritization ranking from the Zonation analysis results.

### 3.2. Zonation Performance Curve Result

Regarding 59 focal species and their habitat distribution maps, the constraint performance curve (Figure 4) generated by Zonation 4 is the graphical representation of the mathematical relationship between the fraction of landscape lost and corresponding remaining biodiversity, meanwhile describing and visualizing conservation priority ranking [47]. Starting from the original intact state of the landscape, the performance curve retains its high-level occurrence of biodiversity features until roughly 40% of the landscape has been lost, which can surely relate to the amount of highly developed urban areas within South Florida. Following that, a slightly steeper shape can be seen on the curve until it reaches approximately 40% of the ranking, depicting the fact that, with small proportions of wildlife-distributed lands lost, almost 90% of species-based biodiversity features can still be preserved. The next noteworthy changing point is that, even after 80% of the landscape has been excluded, the region can keep about 60% of the biodiversity feature compared to the original distribution. Subsequently, the vulnerable ecosystem is estimated to experience dramatic biodiversity degradation once the priority ranking exceeds 80%. Therefore, the top 20% high-priority area and the lowest 40% low-priority area are selected as two thresholds for further analysis. Specifically, the top 20% threshold can be used for conducting conventional spatial conservation prioritization to identify areas with the most ecological significance. The threshold of the lowest 40% is vital for upwards inverse priori-

tization, which would determine the least important sites in terms of biodiversity features, which humans can utilize for future development without disturbing the vulnerable local ecosystem in South Florida. To sum up, South Florida's ecological features are scattered among this region unevenly and the resulting fragmented landscape can be effectively protected by reserving substantial biodiversity-wise sites.

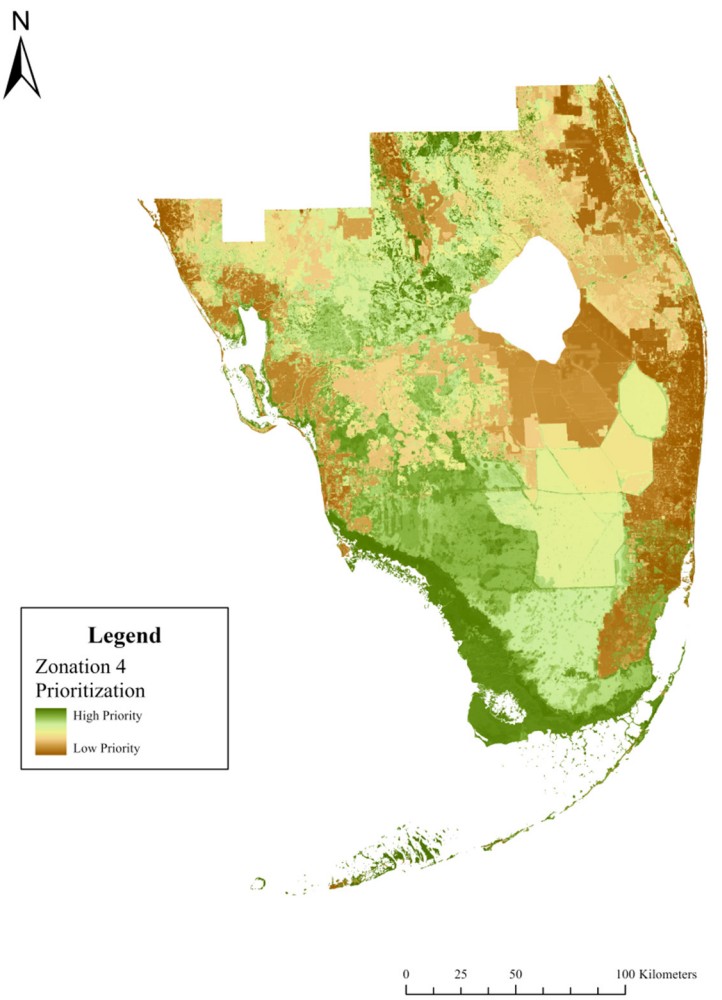

**Figure 3.** Zonation 4 prioritization result of study area.

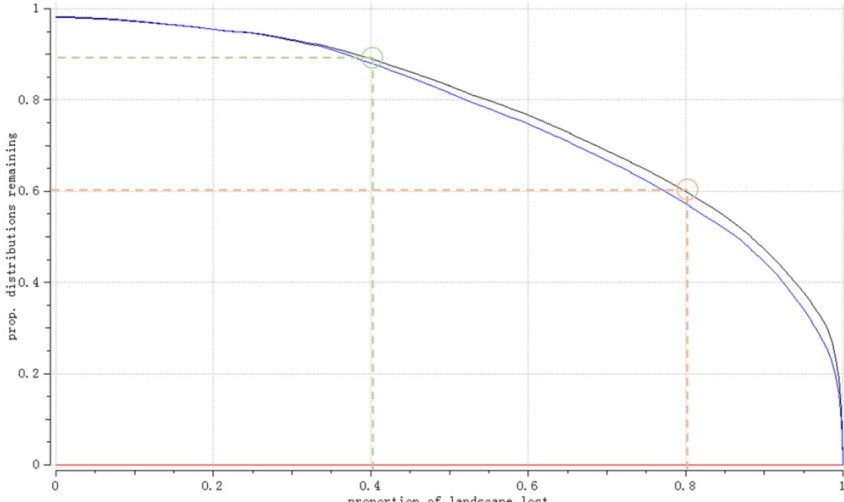

**Figure 4.** Zonation 4 performance curve.

### 3.3. Assessing Florida 2070 Projections

By combining the above Zonation results with Trend and Alternative land-use scenarios from the Florida 2070 project in the ArcGIS platform, we saw two marked turning points on the performance curve as appropriate thresholds, then determined proportions of landscape with the given priority and modeled two sets of assessment maps (Figures 5 and 6) normally and inversely, comparing different conflicting areas of expected future development and ecosystem-wise top 20% priority landscapes, as well as the lowest 40%, which are highlighted in red.

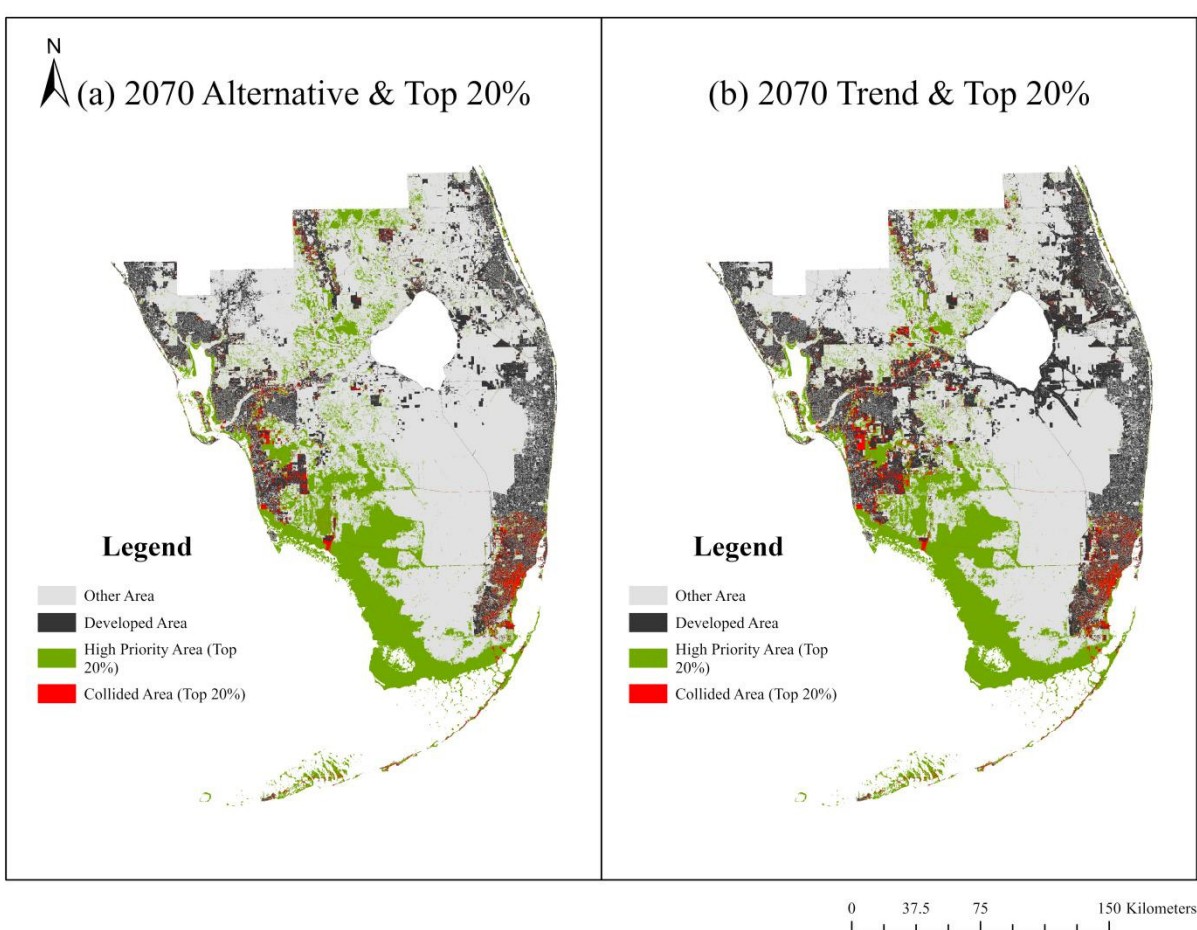

**Figure 5.** Comparison between top 20% prioritization and Florida 2070 Alternative/Trend. (**a**) 2070 Alternative scenario overlapped with the top 20% priority; (**b**) 2070 Trend scenario overlapped with the top 20% priority.

As shown in Figure 5a,b and Table 1, the top 20% priority areas (in green) highly coincided with ecological-substantial areas and are mainly located west of the study area. In the Trend scenario, the developed areas collided with the top 20% priority account for 1169.916 km$^2$, whereas that of the Alternative scenario merely takes up 891.113 km$^2$, which means the alternative plan could prevent 23.8% of ecologically valuable land from being threatened. Spatially, aggressive urban sprawl is expected to consume space around Lake Okeechobee as well as the vast middle-west of the study area, where considerable amounts of spare greenfield sites would be lost. Even though the Alternative can spare 22.3% of greenspace for future exploitation, it is upsetting to apprehend that those conflicting parcels (in red) with high ecological values in the Alternative scenario, which are scattered among south-eastern Florida's highly-urbanized coastal areas, might be inevitable and constant. Acknowledging that reserving the top 20% of the priority area can retain approximately

60% of biodiversity features, the Alternative could consist of only a few more ecologically significant wildlife habitats (+3.6%).

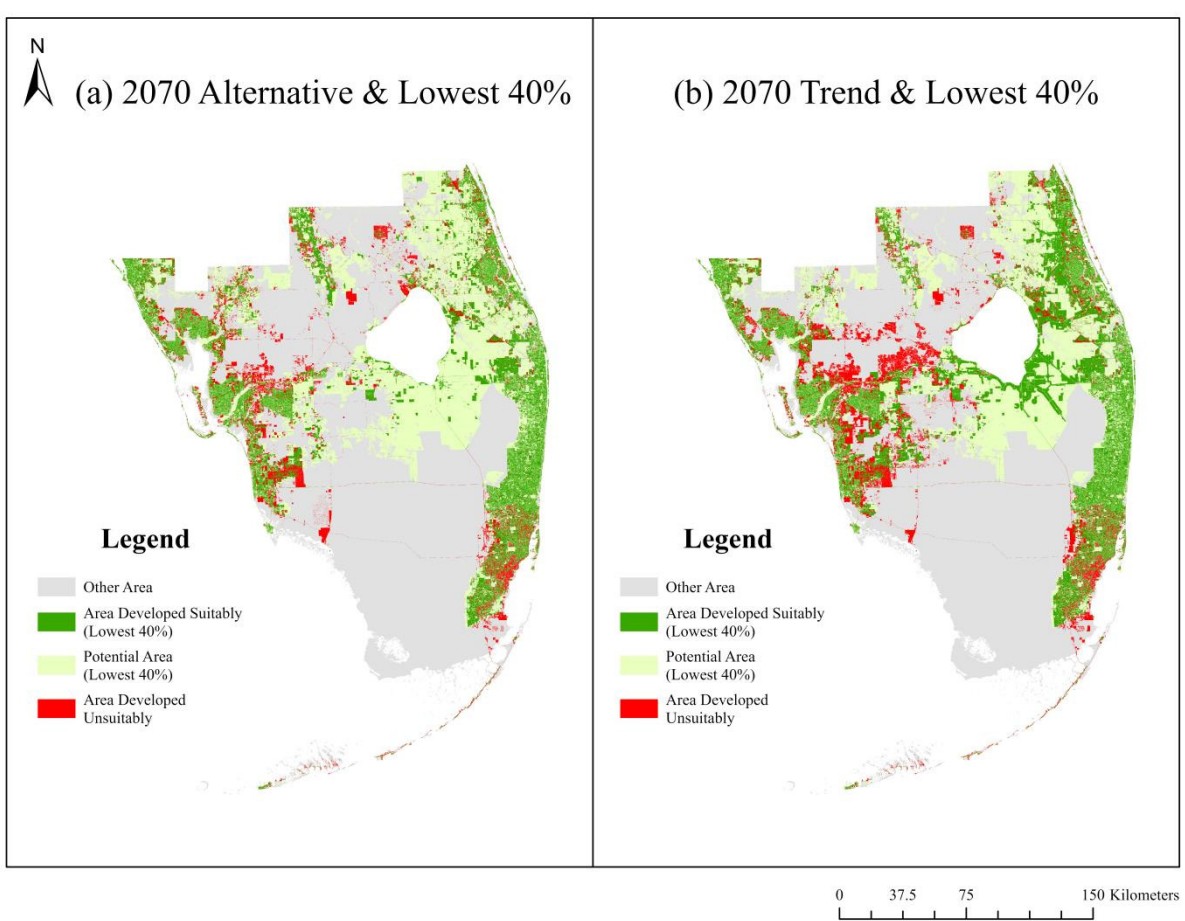

**Figure 6.** Comparison between lowest 40% prioritization and Florida 2070 Alternative/Trend. (**a**) 2070 Alternative scenario overlapped with the lowest 40% priority; (**b**) 2070 Trend scenario overlapped with the lowest 40% priority.

**Table 1.** Related area changes assessed by the top 20% of the ranking.

| Top 20% | Trend | Alternative | Proportion of Changes |
|---|---|---|---|
| Developed area (below top 20%) | 9318.3 km$^2$ | 7143.8 km$^2$ | −22.3% of Trend scenario |
| Collided developed area (top 20%) | 1169.9 km$^2$ | 891.1 km$^2$ | −23.8% of Trend scenario |
| High Priority Area (top 20%) | 7763.8 km$^2$ | 8042.6 km$^2$ | +3.6% of Trend scenario |

From another perspective of upcoming regional development, the assessment of land-use planning in South Florida should also look into potential areas of new construction by evaluating the relationship between biodiversity features and landscape inversely [35]. Based on the Zonation performance curve, the lowest 40% priority cells are of only 9% of the overall ecological value, which are the safest and most suitable places to fulfill upcoming development needs [14]. A significant part of the low-biodiversity lands is naturally distributed within or around metropolitan areas due to anthropogenic influences. Besides, agricultural land is another primary land use type of low-biodiversity landscape [30]. Conflicting areas of inverse assessment maps (Figure 6a,b) imply fields with relatively high value, but which can nevertheless be occupied, and are distributed in a pattern similar to that of overlapped areas in the top 20% assessment (Figure 5a,b). As shown in Table 2, under the Trend scenario, 3209.1 km$^2$ of unsuitable area (in red) are estimated to be taken and developed, while in the Alternative scenario, only 2262.9 km$^2$ of lands would be

developed improperly, assumably protecting nearly 30% of cells from misemployment and risk of degradation. Using the same compact land-use patterns, about 20% of low-richness areas suitable for development and 14.2% of other potential greenspace are planned to be reserved sustainably for the more distant future. To sum up, the conservation effects of the Alternative scenarios are exceptional vis-à-vis the Trend scenario and explicitly address the idea of impact avoidance and the compact city under the inverse assessments of SCP.

**Table 2.** Related area changes assessed by lowest 40% ranking.

| Lowest 40% | Trend | Alternative | Proportion of Changes |
|---|---|---|---|
| Area developed unsuitably (above lowest 40%) | 3209.1 km$^2$ | 2262.9 km$^2$ | −29.5% of Trend scenario |
| Area developed suitably (lowest 40%) | 7279.1 km$^2$ | 5772.0 km$^2$ | −20.1% of Trend scenario |
| Potential area (lowest 40%) | 10,588.3 km$^2$ | 12,095.4 km$^2$ | +14.2% of Trend scenario |

*3.4. Close-Up Assessment and Analysis*

The density of conservation lands in the southwest corner of the Florida peninsula is generally higher than the rest of our study site, which might overlap with highly urbanized areas (such as Lee County mentioned in Section 2.1). Conservation conflicts between human settlements and wildlife habitats can be exaggerated and appear more severe within such places. As Figure 7 shows, the area between Lee County (northwest) and Collier County (southeast) clearly uncovers the differences between two regional land use patterns and their corresponding influences, resulting in violating the territories of top 20% priority. As the business-as-usual development scenario shows in Figure 7b, human beings would possibly conquer adjacent tracts of conservation lands, inevitably leading to habitat loss of Florida panthers, American alligators and other major fauna [48]. On the contrary, following the Alternative compact development pattern, these ecologically significant patches are precisely identified and well reserved in an attempt to combat excessive urban expansion (Figure 7a).

For the sake of sustainability, ecologically less crucial areas should be prioritized for future development rather than occupying other existing greenfield sites. Without the awareness of sustainable development, people tend to exploit land resources at the cost of ecological values. In Figure 8, an example of areas across Desoto County (north) and Charlotte County (south) presents a pronounced contrast between Florida 2070 projections from the viewpoint of inverse conservation prioritization. Figure 8b can be regarded as the ramifications in which the current development pattern might result, where Desoto County's vast acreage of suitable lands for new construction is neglected. However, substantial areas with higher biodiversity value are intensively affected by anthropogenic activities in Charlotte County. To differ from the Trend scenario, an increased proportion of future development does occur within cells with low conservation priority under the Alternative plan (Figure 8a). Unfortunately, some valuable greenfield sites close to the metropolitan border may nonetheless be invaded by humans and likely transformed into suburban and exurban residential areas.

Given the above, close-up assessment and analysis are intended to insinuate detailed planning conflicts no matter whether in the top 20% or lowest 40% priority areas and henceforth unfold the merits and demerits of two scenarios: the ecosystem-wise effectiveness of Alternative Florida 2070 projection could indeed be evaluated and verified by Zonation analysis results; on the other hand, conservation threats and challenges that Trend Florida 2070 projection might bring about are equally unignorable and undeniable. Ranging from identification of valuable lands (protected & agricultural) to excluding them in potential development, these two projection assumptions of the Alternative process explain why the compact development pattern can plausibly contribute more to Florida conservation efforts and in terms of BEBR's population projection, offering a planning-wise resolution for the ongoing human–wildlife conundrum.

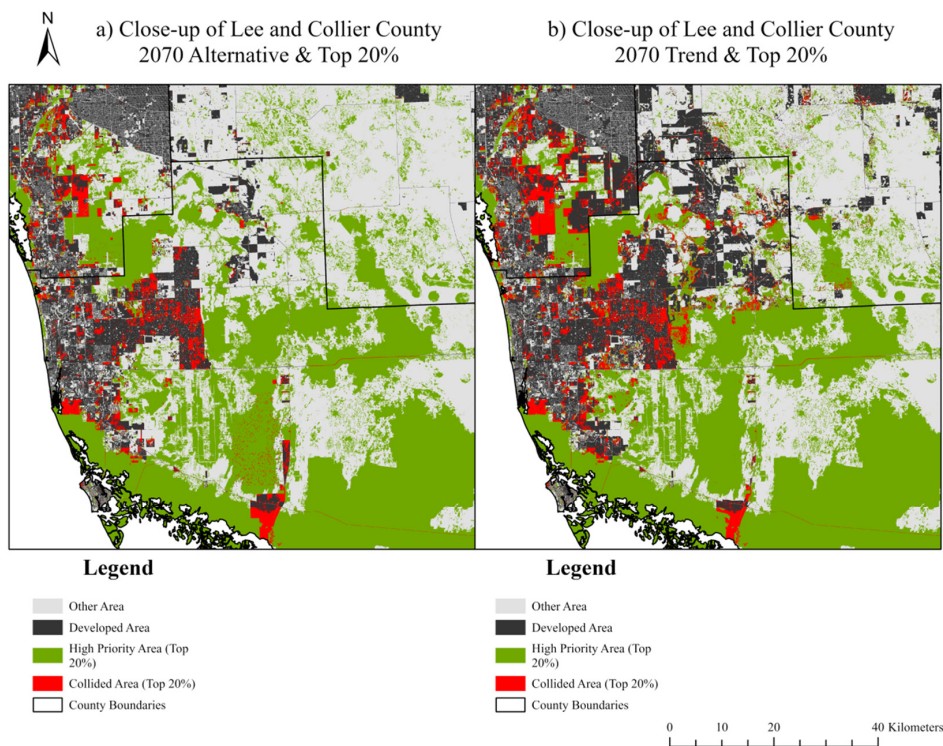

**Figure 7.** Close-comparison between top 20% prioritization and Florida 2070 Trend and Alternative. (**a**) 2070 Alternative scenario overlapped with the top 20% priority; (**b**) 2070 Trend scenario overlapped with the top 20% priority.

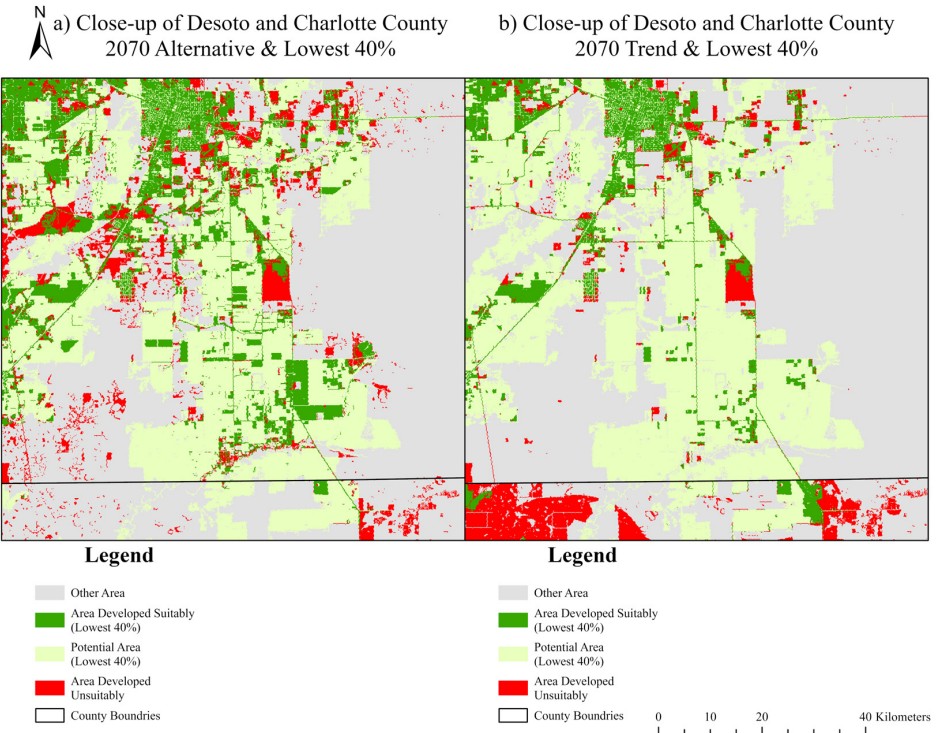

**Figure 8.** Close-comparison between lowest 40% prioritization and Florida 2070 Trend and Alternative. (**a**) 2070 Alternative scenario overlapped with the lowest 40% priority; (**b**) 2070 Trend scenario overlapped with the lowest 40% priority.

## 4. Discussion

The purpose of this study is to compare the human-wise development projections with ecosystem-wise conservation needs in map formats, whereas it historically displays a timeline of Florida's land cover and land use shifts, underscoring current ecologically vital areas along with the highest potential and safest parcels for new constructions. Application of GIS and Zonation 4 software, integrating multi-dimensional aspects of key variables comprehensively, can precisely reveal Florida's struggling situation between wildlife conservation and human development. Based on this analysis, our study serves as a reminder informing Florida conservation efforts about the crucial necessity of balancing or mitigating the conflicts between conservation and anthropogenic activities.

This study can contribute to conservation planning in a multitude of aspects. Firstly, it is unusual to encourage stakeholders to rethink development planning, especially in the Florida 2070 project, from the angle of ecological value and sustainability. Driven by the Florida population estimation in 2070, Trend and Alternative scenarios were generated, implying two distinct land use patterns. Even though the more durable example is recommended, we respect the invaluable environment by effective means such as increasing development densities, as well as protecting reserve areas; all corresponding map projections and strategies were produced and elaborated from a biased perspective of development. Human–wildlife conflicts, as one of the most important subtopics of global biodiversity conservation [2], are widely discussed in the published literature, deeply correlating conservation interventions and conflict management [7,12,22], whereas previous scholars and land-use planners preferably suggested ecosystem-wise spatial planning, rarely addressing intractable ongoing conflicts between biodiversity conservation and social development [19,20]. This study brought in the planning topic of 'Development versus Conservation' and could be referred to as an example of incorporating and considering two sides of human–wildlife conflicts right from the planning stage: the demands of accelerating urban sprawl and prioritization rankings representing the need for biodiversity conservation.

Secondly, we firmly convey ideas on environmental impact avoidance via spatial planning processes for solving conservation conflicts harnessing methodologies of inverse spatial conservation prioritization [35]. Aggregating and constraining human activities to some regions of low ecological merit is the basis of impact avoidance theory; where these impacts need to be prevented are synonymous with the party of environmentally destructive human influences in human–wildlife conflicts [16,35]. To gain more insights into Florida 2070 and its underlying conflicts, our close-up analysis of inverse prioritization assessment is accordingly conducted by comparing conservation ranking maps and development plans. The business-as-usual Trend planning follows Florida's current urban development pattern, aggressively conquering undeveloped greenfield sites regardless of wildlife homeland. In contrast, the Alternative scenario highlights the importance of top-priority areas by delineating ecologically vital areas and encouraging infill and redevelopment [33,34], which reflects the idea of impact avoidance (from the conservation side) and a compact city approach (from the development side) by preferably utilizing land of least biodiversity value but suitable for exploitation, especially around the existing urban fringe (Figure 8a). After all, impact avoidance is convincingly a more effective measure than other conservation prioritization applications [13].

Attention should also be paid to several limitations of the present study. The first issue that needs to be considered and justified is the data and related data process methods underpinning our research. In our study, we substantially employed species data and accordingly relied on the quantitative computational tool (Zonation) generated maps, which should be questioned to some degree. Errors and uncertainty constantly occur throughout different stages of software-assisted conservation planning analysis due to either the quality of underlying data sets or mechanisms of habitat allocation models [11,49,50]. Using static species distribution data as surrogates of existing wildlife occurrences ignores some inherent traits of species activities, such as unpredictable population dynamics and complex

species interactions [21,49,51]. Available species distribution data used here merely act as the role of indicator or subset of the whole population occurring in South Florida [52]. In addition, generally collected via VHF or GPS collars [36,53,54], a species location data downside of hysteresis is intractable and unsolvable, which leads to common dilemmas in planning processes, in that regional biodiversity status or ecological planning issues have not been analyzed and presented to governors or other stakeholders until vital planning strategies and development decisions are grounded [55]. With the purpose of reminding urban planners or decision-makers of Florida's regional ecological value, the result of data analysis, such as the conservation prioritization ranking map in our case, is not sufficiently the best surrogate of Florida's representation of biodiversity resources, acknowledging that there are other alternative conservation planning methods with other concentrations which are also worth being explored (the choice between protecting already endangered species or potentially endangered species) [20].

Concerning human-based population data, Florida 2070 Project's projections laid the foundation for our further overlay analysis, which was incorporated with Zonation South-Florida-species-based results to undertake top-priority and lowest-priority reassessment and inform us of the battle between wildlife conservation and human development shown by the above maps. Even so, 2070 projections are not undoubtedly perfect and to some extent, the predictions would never become a reality. Essentially, urban planning scholars tested and used different assumptions for Trend 2070 and Alternative 2070 regarding different population and development aspects [34]. However, Trend 2070 and Alternative 2070 share the same criteria and weights for examining each cell for determining its suitability for future urban development outwards or infill, which implies the two plans' identical one-sided intentions for human development, rather than regarding conservation value. For sure, the compact scenario would do less harm to the environment by selectively preserving necessary habitats, most of which are constituted by large or connected managed areas, while preferably overlooking the irreplaceable biodiversity importance of small, isolated habitat patches [56]. Besides, the outdated 2010 Baseline is one of the primary data sources for 2070 projections, on which established information mainly includes 2010 gross development density, 2010 land cover pattern and 2010 population distribution. As Figure 9 and Table 3 show, by comparing 2014–2019 land use and cover map (data from Florida Geographic Data Library, University of Florida GeoPlan Center) with 2010 Baseline, the increase in developed area (+61.5%) is beyond astounding in less than twenty years, much higher than the 2070 Trend has expected (+38.3%). Trend 2070 failed to foresee the dramatic infill development that already occurred within urban regions in recent years, which is the plausible main driver of excessive growth in developed lands. Even though Trend 2070 has plotted the extensive conversion of unused land resources to future developed lands, without thoughts of urban redevelopment, the sum of Trend 2070 developed lands (11,735.0 km$^2$) is still less than that of Current 2014–2019 (13,698.0 km$^2$) (in order to present the full map of the study site, the nuances of infill developed land increase might seem implicit in Figure 9). In addition to unplanned area increase, current land use conditions spatially matched the 'leap-frog' urban sprawl pattern [57–59]: human beings have aggressively exceeded the domain of developed lands in Baseline 2010 and even went beyond the estimated future developed area projected in 2070 Trend projection, as well as recklessly infecting harm on several of the most critical reserved areas in South Florida, including the Florida Keys. Unexpected current development conditions and random habitat loss seen below not just uncover underestimations and drawbacks of the Florida 2070 project but also disclose an accelerating pace of landscape conversion, in the sense that actual development pattern and progress should always be deemed sophisticated, dynamic, changeable and unpredictable to some extent. The social preferences reflected by the Florida 2070 project are inevitably restricted by its out-of-date information sources and slightly arbitrary propositions for different land-use patterns.

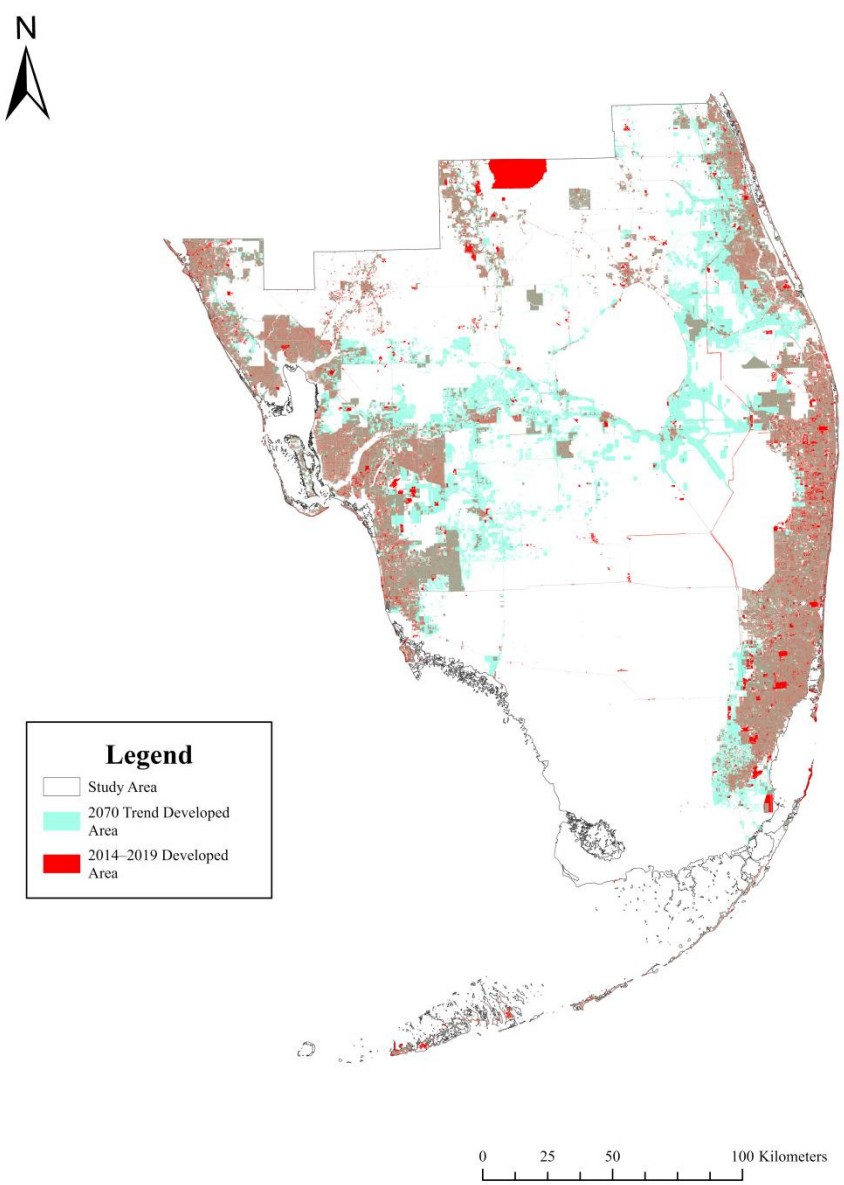

**Figure 9.** Comparison among Florida 2010 Baseline, Florida 2070 Trend and 2014–2019 land-use map.

**Table 3.** Area comparison among Florida 2010 Baseline, Florida 2070 Trend and 2014–2019 land-use map.

| Developed Area | Area | Proportion of Changes |
|---|---|---|
| Baseline 2010 | 8482.3 km$^2$ | Same |
| Current 2014–2019 | 13,698.0 km$^2$ | +61.5% of Baseline 2010 |
| Trend 2070 | 11,735.0 km$^2$ | +38.3% of Baseline 2010 |

Taking Florida as an entire social-ecological system, none of the two dimensions, nature and human, ought to be disregarded since they intricately interact and intertwine, shaping the state's future collectively. Aimed at resolving or mitigating ongoing conservation conflicts involving both human beings and other beings, previous literature suggests that it is imperative for conservation scholars to link natural and social science to create holistic and solid underpinnings for comprehending human–wildlife relationships and impacts [7]. Hence, taking advantage of and integrating data from these two sides would certainly ameliorate our landscape planning strategies, and thus ultimately remediate the disharmony between humans and wildlife [2,7,14]. This paper uniquely links natural values with development preferences [16,25] by analyzing and evaluating the

population-data-based Florida 2070 future scenarios under the matrix by which we assess species-oriented ecological values and prioritize conservation areas spatially. To research the social-ecological "ecology of city" paradigm [23] or study further fundamental elements of human-animal conflict [24] might necessitate more knowledge of the socioeconomic dimension, other than population data, possibly coming from public preferences [60], social expert professional opinions [14], recreational values of sensory perception [61] and particularly vital economic concerns [62]. Over the past decades, a variety of spatial social values analysis methods have been developed and matured [63], including small spatial scale experience mapping [10], sociotope mapping [64] and experience classes (REC-mapping) with different spatial contexts spanning from urban park level identification, or city district level to large-scale regional social feature evaluation [60]. Previously validated processes of gaining information from relevant stakeholders and other social data sources can ensure the availability and credibility of needed human information through approaches like questionnaire surveys, investigation, personal interviews and expert evaluation [60]. Especially, social-economic data may also be involved in the use of spatial conservation prioritization tools, GIS and Marxan, where human-dimensional data can be added as a weighted feature, cost, penalty, or condition layer to see how potential human factors would affect the assessment results [43,65–67]. Initially designed for processing spatial biological data, current SCP tools are not compatible with processing dual dimensions data [25] and may result in emphasizing conservation efforts in lands with high recreational values but low biodiversity richness [8]. Another critical limitation of including more aspects is the arbitrary differentiation and determination of corresponding weightings for data from various categories, which should be tailored and assigned accordingly in relation to deliberate multicriteria approaches involving expert elicitation [8]. Based on the above issues, this study integrated two dimensions by explicitly overlapping and comparing, boldly mapping trade-offs between different land-use needs for city planners. Above all, the relevant reverse design will become much more sophisticated if a later work could be capable of accounting ecological and social criteria discretionarily in a more appropriate manner.

## 5. Conclusions

Not limited to Florida's landscape planning, globally intractable conflicts of conservation versus development ought to be recognized and evaluated in the planning phase of landscape conflict reconciliation [10,68], as we have done here. This study aims to lay a concrete foundation of ecological and social evidence [7,22] to comprehensively reassess plausible development scenarios and inform decision-makers about the significance of restricting human influences when intending to tackle conservation conflicts. By applying the same conflict reassessment and reconciliation framework here, which combines human and natural factors into spatial analysis and accordingly gives out land use suggestions, future relevant scholars and planning activity practitioners can configure and weigh trade-offs within more compendious social-ecological contexts. Nevertheless, like much available literature, our landscape planning analysis project does not include the further succeeding implementations and eventual enhancements of [69], inevitably leaving cognitive inconsistencies between methodological studies and practical effectiveness in such conservation activities. Acknowledging procedural shortcomings in our present studies, we sincerely recommend including sufficient evidence of outcomes in future conservation planning work.

**Author Contributions:** Conceptualization, F.L. and M.Z.; methodology, F.L., M.Z. and F.C.; software, F.L. and F.C.; validation, F.L.; formal analysis, F.L.; investigation, F.L. and F.C.; resources, M.Z.; data curation, F.L. and F.C.; writing—original draft preparation, F.L.; writing—review and editing, M.Z. and F.L.; visualization, F.L.; supervision, M.Z.; project administration, M.Z. All authors have read and agreed to the published version of the manuscript.

**Funding:** This research was founded by Guangdong Natural Science Foundation (2020A1515011072). The APC was funded by Guangdong Natural Science Foundation (2020A1515011072).

**Data Availability Statement:** [FLORIDA 2070 PROJECT: 2010 BASE SCENARIO DEVELOPED LANDS] University of Florida GeoPlan Center. 2017. FLORIDA 2070 PROJECT: 2010 BASE SCENARIO DEVELOPED LANDS; Florida Geographic Data Library. [FLORIDA 2070 PROJECT: 2010 BASE SCENARIO PROTECTED LANDS] University of Florida GeoPlan Center. 2017. FLORIDA 2070 PROJECT: 2010 BASE SCENARIO PROTECTED LANDS; Florida Geographic Data Library. [FLORIDA 2070 PROJECT: 2070 TREND SCENARIO DEVELOPED LANDS] University of Florida GeoPlan Center. 2017. FLORIDA 2070 PROJECT: 2070 TREND SCENARIO DEVELOPED LANDS; Florida Geographic Data Library. [FLORIDA 2070 PROJECT: 2070 TREND SCENARIO PROTECTED LANDS] University of Florida GeoPlan Center. 2017. FLORIDA 2070 PROJECT: 2070 TREND SCENARIO PROTECTED LANDS; Florida Geographic Data Library. [FLORIDA 2070 PROJECT: 2070 ALTERNATIVE SCENARIO DEVELOPED LANDS] University of Florida GeoPlan Center. 2017. FLORIDA 2070 PROJECT: 2070 ALTERNATIVE SCENARIO DEVELOPED LANDS; Florida Geographic Data Library. [FLORIDA 2070 PROJECT: 2070 ALTERNATIVE SCENARIO PROTECTED LANDS] University of Florida GeoPlan Center. 2017. FLORIDA 2070 PROJECT: 2070 ALTERNATIVE SCENARIO PROTECTED LANDS; Florida Geographic Data Library. [Land Use and Cover (FLUCCS Level 3) by Water Management District in Florida 2014–2019] University of Florida GeoPlan Center. 2017. Land Use and Cover (FLUCCS Level 3) by Water Management District in Florida 2014–2019; Florida Geographic Data Library. U.S. Geological Survey (USGS)—Gap Analysis Project (GAP), 2018, U.S. Geological Survey—Gap Analysis Project Species Habitat Maps CONUS_2001: U.S. Geological Survey data release, https://doi.org/10.5066/F7V122T2 (accessed on 2 November 2022). NatureServe. 2021. NatureServe Network Biodiversity Location Data accessed through NatureServe Explorer [web application]. NatureServe, Arlington, Virginia. Available https://explorer.natureserve.org/ (accessed on 10 November 2021).

**Acknowledgments:** We thank Design School, South China University of Technology for technical computational supports and thank Graduate Writing Center, UC Berkeley for writing suggestions.

**Conflicts of Interest:** The authors declare no conflict of interest.

## Appendix A

**Table A1.** South Florida focal species list.

| Common Name | Scientific Name | Type | Global Rank * | State Rank ** |
|---|---|---|---|---|
| American Alligator | Alligator mississippiensis | Amphibian and reptile | G5 | S4 |
| American crocodile | Crocodylus acutus | Amphibian and reptile | G2 | S2 |
| American oystercatcher | Haematopus palliates | Bird | G5 | S2 |
| Atlantic salt marsh snake | Nerodia clarkii taeniata | Amphibian and reptile | T1 | S1 |
| Audubon's crested caracara | Polyborus plancus audubonii | Bird | G5 | S2 |
| Bald eagle | Haliaeetus leucocephalus | Bird | G5 | S3 |
| Big Cypress fox squirrel | Sciurus niger avicennia | Mammal | T2 | S2 |
| Black rail | Laterallus jamaicensis | Bird | G3 | S2 |
| Black skimmer | Black skimmer | Bird | G5 | S3 |
| Black-whiskered vireo | Vireo altiloquus | Bird | G5 | S3 |
| Bluetail mole skink | Plestiodon egregius lividus | Amphibian and reptile | T2 | S2 |
| Cape Sable seaside sparrow | Ammodramus maritimus mirabilis | Bird | T1 | S1 |
| Cooper's Hawk | Accipiter cooperii | Bird | G5 | S3 |
| Eastern diamondback rattlesnake | Crotalus adamanteus | Amphibian and reptile | G3 | S3 |
| Eastern indigo snake | Drymarchon couperi | Amphibian and reptile | G3 | S2 |

**Table A1.** *Cont.*

| Common Name | Scientific Name | Type | Global Rank * | State Rank ** |
|---|---|---|---|---|
| Everglades mink | Neovison vison evergladensis | Mammal | G5 | S2 |
| Florida black bear | Ursus americanus floridanus | Mammal | T4 | S4 |
| Florida bonneted bat | Eumops floridanus | Mammal | G1 | S1 |
| Florida burrowing owl | Athene cunicularia | Bird | G4 | S3 |
| Florida grasshopper sparrow | Ammodramus savannarum floridanus | Bird | G5 | S1 |
| Florida Key deer | Odocoileus virginianus clavium | Mammal | T1 | S1 |
| Florida Keys mole skink | Plestiodon egregius egregius | Amphibian and reptile | T1 | S1 |
| Florida panther | Puma concolor coryi | Mammal | G5 | S1 |
| Florida sandhill crane | Antigone canadensis pratensis | Bird | G5 | S2 |
| Florida scrub lizard | Sceloporus woodi | Amphibian and reptile | G2 | S2 |
| Florida scrub-jay | Aphelocoma coerulescens | Bird | G2 | S2 |
| Gopher tortoise | Gopherus polyphemus | Amphibian and reptile | G3 | S3 |
| Key Largo woodrat | Neotoma floridana smalli | Mammal | T1 | S1 |
| Least tern | Sternula antillarum | Bird | G4 | S3 |
| Limpkin | Aramus guarauna | Bird | G5 | S3 |
| Little blue heron | Egretta caerulea | Bird | G5 | S4 |
| Lower Keys rabbit | Sylvilagus palustris hefneri | Mammal | T1 | S1 |
| Mangrove cuckoo | Coccyzus minor | Bird | G5 | S3 |
| Marsh rabbit | Sylvilagus palustris hefneri | Mammal | G5 | S5 |
| Mottled duck | Anas fulvigula | Bird | G4 | S3 |
| Ornate diamondback terrapin | Malaclemys terrapin macrospilota | Amphibian and reptile | T4 | N/A |
| Piping plover | Charadrius melodus | Bird | G3 | S2 |
| Red-cockaded woodpecker | Picoides borealis | Bird | G3 | S2 |
| Reddish egret | Egretta rufescens | Bird | G4 | S2 |
| Rice rat | Oryzomys palustris natator | Mammal | T2 | S2 |
| Rim rock crowned snake | Tantilla oolitica | Amphibian and reptile | G1 | S1 |
| roseate spoonbill | Platalea ajaja | Bird | G5 | S2 |
| Roseate tern | Sterna dougallii dougallii | Bird | T3 | S1 |
| Rufa red knot | Calidris canutus rufa | Bird | T2 | S2 |
| Sand skink | Plestiodon reynoldsi | Amphibian and reptile | G3 | S3 |
| Sherman's fox squirrel | Sciurus niger shermani | Mammal | G5 | T3 |
| Short-tailed hawk | Buteo brachyurus | Bird | G4 | S1 |
| Snail kite | Rostrhamus sociabilis | Bird | G4 | S2 |
| Snowy egret | Egretta thula | Bird | G5 | S3 |
| Snowy plover | Charadrius nivosus | Bird | G3 | S1 |
| Southeastern American kestrel | Falco sparverius paulus | Bird | G5 | S3 |

**Table A1.** *Cont.*

| Common Name | Scientific Name | Type | Global Rank * | State Rank ** |
|---|---|---|---|---|
| Southern chorus frog | Pseudacris nigrita | Amphibian and reptile | G5 | S5 |
| Striped mud turtle | Striped mud turtle | Amphibian and reptile | G5 | S5 |
| Swallow-tailed kite | Elanoides forficatus | Bird | G5 | S2 |
| Tricolored heron | Egretta tricolor | Bird | G5 | S4 |
| White crowned pigeon | Patagioenas leucocephala | Bird | G3 | S3 |
| white ibis | Eudocimus albus | Bird | G5 | S4 |
| Whooping crane | Grus americana | Bird | G1 | N/A |
| Wood stork | Mycteria Americana | Bird | G4 | S2 |

* NatureServe global conservation status ranks include G1 (critically imperiled), G2 (imperiled), G3 (vulnerable), G4 (apparently secure) and G5 (secure). Especially for infraspecific taxon (subspecies or varieties), the global conservation status are indicated by a "T-rank" following the species' global rank, like T1, T2, T3, T4 and T5. ** NatureServe subnational state conservation status ranks include S1 (critically imperiled), S2 (imperiled), S3 (vulnerable), S4 (apparently secure) and S5 (secure).

**Table A2.** The $\alpha$ value for focal species in South Florida.

| Common Name | Weighting | Dispersal Capability | $\alpha$Value in Zonation ($\alpha$ = 2/Dispersal Capability) |
|---|---|---|---|
| American Alligator | 2 | 15,000 | 0.000130 |
| American crocodile | 4 | 30,000 | 0.000067 |
| American oystercatcher | 4 | 5000 | 0.00040 |
| Atlantic salt marsh snake | 5 | 10,000 | 0.00020 |
| Audubon's crested caracara | 4 | 20,000 | 0.00010 |
| Bald eagle | 3 | 10,000 | 0.00020 |
| Big Cypress fox squirrel | 4 | 5000 | 0.00040 |
| Black rail | 4 | 5000 | 0.00040 |
| Black skimmer | 3 | 5000 | 0.00040 |
| Black-whiskered vireo | 3 | 5000 | 0.00040 |
| Bluetail mole skink | 4 | 5000 | 0.00040 |
| Cape Sable seaside sparrow | 5 | 5000 | 0.00040 |
| Cooper's Hawk | 3 | 10,000 | 0.00020 |
| Eastern diamondback rattlesnake | 3 | 5000 | 0.00040 |
| Eastern indigo snake | 4 | 10,000 | 0.00020 |
| Everglades mink | 4 | 790 | 0.00253 |
| Florida black bear | 2 | 150,000 | 0.0000133 |
| Florida bonneted bat | 5 | 5000 | 0.00040 |
| Florida burrowing owl | 3 | 5000 | 0.00040 |
| Florida grasshopper sparrow | 5 | 5000 | 0.00040 |
| Florida Key deer | 5 | 2600 | 0.00077 |
| Florida Keys mole skink | 5 | 5000 | 0.00040 |
| Florida panther | 5 | 29,000 | 0.000069 |
| Florida sandhill crane | 4 | 15,000 | 0.00013 |
| Florida scrub lizard | 4 | 5000 | 0.00040 |
| Florida scrub-jay | 4 | 3500 | 0.00057 |
| Gopher tortoise | 3 | 4000 | 0.00050 |
| Key Largo woodrat | 5 | 5000 | 0.00040 |
| Least tern | 3 | 5000 | 0.00040 |
| Limpkin | 3 | 5000 | 0.00040 |
| Little blue heron | 2 | 10,000 | 0.00020 |
| Lower Keys rabbit | 5 | 10,000 | 0.00020 |

**Table A2.** *Cont.*

| Common Name | Weighting | Dispersal Capability | $\alpha$Value in Zonation ($\alpha$ = 2/Dispersal Capability) |
|---|---|---|---|
| Mangrove cuckoo | 3 | 5000 | 0.00040 |
| Marsh rabbit | 1 | 10,000 | 0.00020 |
| Mottled duck | 3 | 10,000 | 0.00020 |
| Ornate diamondback terrapin | 1 | 10,000 | 0.00020 |
| Piping plover | 4 | 5000 | 0.00040 |
| Red-cockaded woodpecker | 4 | 8000 | 0.00025 |
| Reddish egret | 4 | 10,000 | 0.00020 |
| Rice rat | 4 | 5000 | 0.00040 |
| Rim rock crowned snake | 5 | 5000 | 0.00040 |
| roseate spoonbill | 4 | 10,000 | 0.00020 |
| Roseate tern | 5 | 5000 | 0.00040 |
| Rufa red knot | 4 | 5000 | 0.00040 |
| Sand skink | 3 | 5000 | 0.00040 |
| Sherman's fox squirrel | 3 | 740 | 0.00270 |
| Short-tailed hawk | 5 | 10,000 | 0.00020 |
| Snail kite | 4 | 10,000 | 0.00020 |
| Snowy egret | 3 | 10,000 | 0.00020 |
| Snowy plover | 5 | 1000 | 0.00200 |
| Southeastern American kestrel | 3 | 10,000 | 0.00020 |
| Southern chorus frog | 1 | 5000 | 0.00040 |
| Striped mud turtle | 1 | 10,000 | 0.00020 |
| Swallow-tailed kite | 4 | 10,000 | 0.00020 |
| Tricolored heron | 2 | 10,000 | 0.00020 |
| White crowned pigeon | 3 | 10,000 | 0.00020 |
| white ibis | 2 | 10,000 | 0.00020 |
| Whooping crane | 5 | 5000 | 0.00040 |
| Wood stork | 4 | 10,000 | 0.00020 |

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
