# Peer review of "Conservation and Development: Reassessing the Florida 2070 Planning Project with Spatial Conservation Prioritization"

_land, doi:10.3390/land11122182_

Round 1

Reviewer 1 Report

The manuscript using land vertebrate distribution data to assess the different land use planning project in south florida. the result clearly showed that the alternative plan better protect the animal habitat than the trend plan. As biodiversity conservation becomes more and more important in economic develepment and human welfare, the manuscript gave us a mature method to evualuate how takes into account wildlife when plans land use. The manuscript are well written in proper way. I have  some suggestions for the authors.

1. the manuscript is somewhat too long. I am sure authors can shorten the manuscript without weakening what authors wanna express.

2. the section materials and methods should be more accurate. in the line 267, the abbreviation GUI have to be mentioned before used.  the figure legend developed area, collied area, potential area etc. should be defined in the methods. how 59 species were selected should be clarified, for most endangered or most abundant? why authors selected  two counties  to repeat the anylysis and why choosed these two counties should also be  clarified. 

Reviewer 2 Report

Stimulating study. Well written. Clear methodology.

Limitations are fully discussed.

I suggest accept with minor changes.

L105: I suggest the use of square km instead of acres, like in the rest of document.    

Figure 5 and other figures: Use km instead of meters in map scales. A lot of zeros.

Table 1. Table 2… specify units… square km?

L337:   23.8%  … specify “ of trend scenario “…

Figure 5. captions mentions A and B in capital letters, while in text referred as not capital letters.

Figure 6. Idem. C and D are mentioned.

What is the resolution or cell size ? scale of study should be mentioned at some point. 

L376: exexplicitly

L401: Desoto County and Charlotte County boundary is not seen in map of figure 8.

L514: The inclusion of figure 9 is very interesting for the discussion, maybe give a few numbers, itcould help quantify these issues, as done in result section…

Appendix A. Explain o mention reference of Global rank and State rank values…. G5 ? S4?

L633, 634. Words repeated. Check PDF

Reviewer 3 Report

The authors in this paper propose an analysis of spatial conservation priorities based on focal species using Zonation 4 software aimed at identifying the most significant areas and evaluating two different land use scenarios provided by the Florida 2070 project.

The paper is interesting.

The paper overall is well structured. It is suggested to provide a more in-depth presentation of the techniques of inverse spatial conservation prioritization.

It is suggested to provide some information on how the Zonation 4 software provides the information, for those unfamiliar with the software it is difficult to identify the input and output data, or the robustness checks of the results.

It is suggested to put part of the text before the beginning of subsections 3.3. and 3.4, i.e., to avoid starting the subsection directly with the figure.

It is suggested to present the two scenarios 2070 Trend and 2070 Alternative, what are for example the differences in terms of urbanized land development , what are the planned interventions etc.
